# Synergistic Effects of Bacteriophage vB_Eco4-M7 and Selected Antibiotics on the Biofilm Formed by Shiga Toxin-Producing *Escherichia coli*

**DOI:** 10.3390/antibiotics11060712

**Published:** 2022-05-25

**Authors:** Agnieszka Necel, Sylwia Bloch, Gracja Topka-Bielecka, Agata Janiszewska, Aleksandra Łukasiak, Bożena Nejman-Faleńczyk, Grzegorz Węgrzyn

**Affiliations:** Department of Molecular Biology, Faculty of Biology, University of Gdansk, Wita Stwosza 59, 80-308 Gdansk, Poland; agnieszka.necel@phdstud.ug.edu.pl (A.N.); sylwia.bloch@ug.edu.pl (S.B.); gracja.topka-bielecka@ug.edu.pl (G.T.-B.); a.janiszewska.477@studms.ug.edu.pl (A.J.); a.lukasiak.639@studms.ug.edu.pl (A.Ł.)

**Keywords:** ciprofloxacin, rifampicin, phage therapy, Shiga toxin-producing *E. coli*, biofilm

## Abstract

Apart from antibiotic resistance of pathogenic bacteria, the formation of biofilms is a feature that makes bacterial infections especially difficulty to treat. Shiga toxin-producing *Escherichia coli* (STEC) strains are dangerous pathogens, causing severe infections in humans, and capable of biofilm production. We have reported previously the identification and characterization of the vB_Eco4-M7 bacteriophage, infecting various STEC strains. It was suggested that this phage might be potentially used in phage therapy against these bacteria. Here, we tested the effects of vB_Eco4-M7 alone or in a phage cocktail with another STEC-infecting phage, and/or in a combination with different antibiotics (ciprofloxacin and rifampicin) on biofilm formed by a model STEC strain, named *E. coli* O157:H7 (ST2-8624). The vB_Eco4-M7 phage appeared effective in anti-biofilm action in all these experimental conditions (2–3-fold reduction of the biofilm density, and 2–3 orders of magnitude reduction of the number of bacterial cells). However, the highest efficiency in reducing a biofilm’s density and number of bacterial cells was observed when phage infection preceded antibiotic treatment (6-fold reduction of the biofilm density, and 5–6 orders of magnitude reduction of the number of bacterial cells). Previous reports indicated that the use of antibiotics to treat STEC-caused infections might be dangerous due to the induction of Shiga toxin-converting prophages from bacterial genomes under stress conditions caused by antibacterial agents. We found that ciprofloxacin was almost as efficient in inducing prophages from the *E. coli* O15:H7 (ST2-8624) genome as a classical inducer, mitomycin C, while no detectable prophage induction could be observed in rifampicin-treated STEC cells. Therefore, we conclude the latter antibiotic or similarly acting compounds might be candidate(s) as effective and safe drug(s) when used in combination with phage therapy to combat STEC-mediated infections.

## 1. Introduction

Shiga toxin-producing *Escherichia coli* (STEC) strains are pathogens causing various symptoms in humans, from bloody diarrhea to severe complications, like hemolytic uremic syndrome which is especially dangerous and leads to the death of over 10% of patients [1,2,3]. The high level of virulence of STEC depends mostly on the production of Shiga toxin which displays enzymatic activity causing inactivation of 28S rRNA (and thus, inactivation of the whole eukaryotic ribosome) by removing a single adenine residue in this nucleic acid [4]. Interestingly, Shiga toxins are encoded by the *stx* genes, located on lambdoid bacteriophages which are present in STEC genomes in the form of prophages, called Shiga toxin-converting phages, or Stx phages [5]. Since effective expression of the vast majority of genes present in prophage DNA is possible only after prophage induction, such a process must occur to activate the virulence of STEC (only negligible expression of *stx* genes is possible in lysogenic cells) [5,6,7]. In the human gut, the prophage inducers include reactive oxygen species and other factors causing the S.O.S. response in bacterial cells [8]. Unfortunately, some antibiotics can also induce Stx prophages which is especially problematic in the treatment of STEC-infected patients. The antibiotic-mediated prophage induction, despite killing bacteria, can also lead to massive production of Shiga toxins and subsequent patient death [9,10]. Therefore, the use of these therapeutics must be very cautious when STEC infection is confirmed or even suspected [11].

Another problem with the treatment of bacterial infections, including those caused by STEC, is the formation of biofilms. These structures consist of microorganisms which are attached to any surface through and surrounded by an extracellular exopolysaccharide matrix. Microbial cells are submerged in a mixture of various compounds, like proteins, lipids, polysaccharides, and nucleic acids, described as extracellular polymeric substances [12]. Such a structure of the biofilm protects microbial cells against different chemical, physical, and biological factors. In fact, penetration of biofilms by antibiotics is severely impaired relative to conditions found in bacterial liquid cultures, on nutrient plates, or in bodies of patients if planktonic forms of microbes predominate [13].

The problems presented above with effective treatment of STEC infections indicate the need for developing alternative therapeutic procedures. One possibility is phage therapy, based on the use of bacteriophages which can infect pathogenic bacteria and develop lytically in their cells, causing the death of these microorganisms [14]. In this light, in our previous studies, we have isolated and characterized a bacteriophage that is specific to *E. coli* O157, the most frequent serotype among STEC strains. This phage, named vB_Eco4M-7, belongs to the *Myoviridae* family, displays a lytic mode of development, does not encode any known toxins, and does not cause induction of prophages in infected STEC cells [15]. This phage appeared to be effective in the protection of vegetables from contamination with STEC, and it was demonstrated to be non-toxic for mammalian cells [16]. Since other bacteriophages were reported previously to be effective in destroying bacterial cells included in biofilms [17,18,19], in this work, we have tested the effects of vB_Eco4M-7 used alone or in combination with another STEC-infecting phage, ECML-117 (forming together the phage cocktail) and/or different antibiotics (ciprofloxacin and rifampicin) on biofilm formed by STEC.

## 2. Results

In our experiments, we have employed the vB_Eco4M-7 bacteriophage [15,16], and tested its effects on biofilms formed by a model STEC strain, named *E. coli* O157:H7 (ST2-8624) [15]. To prepare a bacteriophage cocktail, vB_Eco4M-7 was mixed with another STEC-infecting phage, ECML-117 which has been broadly investigated by other scientists previously, as a component of the phage cocktail [20,21] and by us, in a comparative study with vB_Eco4M-7 [15]. These experiments indicated that both phages belong to the *Myoviridae* family, and showed the same morphology of virions and plaques, comparable resistance to laboratory disinfectants, almost identical rate of adsorption to the *E. coli* O157:H7 (ST2-8624) host, and similar lytic development with short eclipse and latent periods and burst size of approximately 100 phage particles per cell. On the other hand, when monitoring the infection process of *E. coli* O157:H7 (ST2-8624) cells, the vB_Eco4M-7 phage was more efficient in killing bacteria than ECML-117 [15]. For this reason and for clarity of the presented results, we have decided to focus on the phage vb_Eco4M-7, isolated and characterized previously by us [15,16].

Three different antibiotics, presenting various mechanisms of action, were also used (Table 1). Ciprofloxacin and rifampicin were tested individually or in mixtures with bacteriophage(s) against bacterial biofilms. Additionally, these antibiotics and mitomycin C (a positive control) were used in experiments performed to test prophage induction in the *E. coli* O157:H7 (ST2-8624) strain. Characteristics of these antibiotics, including their modes of action on bacterial cells, are presented in Table 1. We have estimated minimal inhibitory concentrations (MIC) for all these antibiotics against the *E. coli* O157:H7 (ST2-8624) strain. Bacteria were grown in the LB medium supplemented with various concentrations of tested antibiotics for 24 h. Measurement of OD_600_ indicated specific thresholds in the concentrations causing growth inhibition for all these antimicrobial agents. Formally, the MIC value was determined as an antibiotic concentration causing severe inhibition of the bacterial culture growth after 24 h incubation as measured by OD_600_ < 0.05. The calculated values are presented in Table 1, and the experimental results of measurements (allowing such calculations) are shown in Appendix A. The MIC value for ciprofloxacin was significantly lower (0.03 mg/mL) than the clinical breakpoint for this antibiotic in the treatment of Enterobacterales, estimated as 0.25 mg/mL; a breakpoint for rifampicin in this group of bacteria has not been established (https://www.eucast.org/fileadmin/src/media/PDFs/EUCAST_files/Breakpoint_tables/v_12.0_Breakpoint_Tables.pdf (accessed on 22 May 2022)). We have considered MIC values rather than MBC-B, as inhibition of the bacterial growth appears more important for the practical use (preventing multiplication of pathogenic cells) than actual killing of the bacterial cells.

To test the effects of bacteriophage vB_Eco4M-7 on biofilms formed by the *E. coli* O157:H7 (ST2-8624) strain, we have cultured bacteria under conditions allowing biofilm production [26]. Biofilms were formed on 12-well polystyrene microtiter plates for 48 h at 37 °C without shaking. After incubation, the liquid parts were discarded and the mature biofilms were washed once with PBS buffer to remove loosely adherent cells.

Phage lysate was applied to the biofilm-bearing wells to an MOI of 100 (10^10^ PFU/well). A similar strategy to combat *E. coli* biofilms was also presented previously [27]. The vB_Eco4M-7 phage was used either alone or in the cocktail (this term is used further in the text). The plates were statically incubated for an additional 6 h at 37 °C in the presence of an antimicrobial agent. Afterward, the mixtures were discarded and the biofilms were treated once with PBS buffer to remove the unattached planktonic *E. coli* cells.

To evaluate the antibiofilm properties of tested phages or the phage cocktail, the following assays were used: measurement of OD_600_, estimation of the number of living bacterial cells by determining colony forming units per ml (CFU/mL), quantification of the biofilm biomass through densitometric measurement and crystal violet (CV) staining. The biofilm cell density and the number of bacteria adhered to the surface of microplate wells were analyzed after the suspension of the samples in the PBS buffer by vigorous pipetting. In turn, the results obtained after biofilm staining with CV were confirmed by the densitometric method based on the quantification of the pixel density of each well with the fixed biofilm layer that correlated to the thickness of the total biofilm biomass.

Using these methods, we determined that vB_Eco4M-7 is effective in reducing STEC-formed biofilm as the optical density of the culture, number of viable bacterial cells, and density of the biofilm were significantly lower in samples incubated in the presence of this bacteriophage relative to the control (untreated biofilm) obtained after additional 6 h incubation at 37 °C (Figure 1). The effects were even more pronounced when the phage cocktail was used instead of the vB_Eco4M-7 lysate, indicating that a mixture of different phages can be especially effective in destroying biofilms formed by STEC (Figure 1).

Furthermore, statistical analysis indicated that the reduction of the optical density (OD_600_) of suspended bacterial biofilm after 6 h incubation with antimicrobial agents was also statistically significant (the one-way ANOVA with *p* value < 0.05) compared to the bacterial density of the non-treated control variant measured at 48 h of biofilm maturity (data not shown). Nevertheless, a fraction of cells surviving in the biofilm in the presence of bacteriophages was still significant. Therefore, in the next step of our studies, we assessed the efficiency of mixtures of bacteriophages and antibiotics in combating bacterial biofilms produced by the *E. coli* O157:H7 (ST2-8624) strain.

To test the effects of mixtures of phages and antibiotics on biofilms, the experiments were performed according to the diagram presented in Figure 2. Biofilms were formed by incubation for 48 h, and then they were treated with either tested agents (vB_Eco4M-7, the phage cocktail or antibiotic) individually, sequentially, or simultaneously. Similar to other analyzed agents, the activity of the vB_Eco4M-7 phage alone against *E. coli* O157:H7 (ST2-8624) bacteria in biofilm was tested after 6 h. Indeed, in the preliminary experiment, we tested the impact of vB_Eco4M-7 phage alone on biofilm also in shorter time treatments, after 2 and 4 h, and with a comparison to control we observed a significant decrease in OD value just after 2 h of incubation with this phage (Appendix A, *p* < 0.001). Importantly, in comparison to 2 h-long treatment, the effects observed after 4 and 6 h were even more pronounced (Appendix A). Therefore, we made a decision to run the whole experiment for 6 h. Our intention of conducting experiments with washing phages out after 3 h incubation was to determine if potential improved efficacy of antibiotic treatment may be due to the facilitation made by actions of bacteriophages. After further incubation, the following parameters were measured: viability of bacterial cells, optical density of the culture, density of the biofilm, antibiotic resistance of survivors, sensitivity of survivors to phages, and efficiency of prophage induction from the bacterial host genome. Details of experimental schemes are shown in Figure 2.

When comparing the effects of phages and antibiotics on the reduction of STEC biofilms, we found that although bacteriophages and antibiotics revealed some efficiency when used alone, the highest efficacy in decreasing both biofilm density and number of living bacterial cells could be observed when a combination of these antibacterial agents was tested. Concerning OD_600_ analysis of *E. coli* biofilm, all tested antimicrobial agents alone or in combination showed a statistically significant biofilm density reduction relative to the controls prior to (data not shown) and after additional 6 h incubation at 37 °C. Among these combinations, the phage cocktail was more efficient than a single phage, while the highest efficiency was evident in experiments where phages were applied first, and antibiotic was added subsequently. These rules applied to both antibiotics, ciprofloxacin, and rifampicin. In addition, we performed an analysis to determine whether the nature of the mixed treatments of phage vB_Eco4M-7 alone or in a cocktail and tested antibiotics (rifampicin and ciprofloxacin) against *E. coli* O157:H7 (ST2-8624) biofilm is synergistic. The variants which showed such an interaction were marked in Figure 3 by “S” (for synergy). Importantly, the effect of the sequential treatment for phage vB_Eco4M-7 alone added prior to rifampicin, and phage Eco4M-7 in the cocktail added before rifampicin or ciprofloxacin were synergistic (Figure 3). The calculations were made according to the formula developed previously by others [28] (Figure 3).

To confirm that the synergistic effect was the result of the actual interactions of phages with the antibiotic rather than arising from getting rid of the medium and putative partial disruption of the biofilm structure, we have applied an additional sequential control variant of the experiment. In this variant, the biofilm was first treated for 3 h with a phage-free medium, and then the medium was removed and replaced with the tested antibiotic (ciprofloxacin or rifampicin). Similarly to the other tested sequential variants (presented in Figure 3), in this control, no additional step, such as washing or pelleting, was introduced after the first stage of the experiment which was the removal of the phage-free medium. When the results of the sequential action of phages and antibiotics were compared to this new sequential control, the effects of the treatments in which phage vB_Eco4M-7 alone or in the phage cocktail was added prior to antibiotics were still synergistic and showed a statistically significant reduction in bacterial biofilm density measured at OD_600_, and in the number of surviving cells (Appendix A). This indicated that the removal of the medium after the first stage of the sequential treatment did not disrupt the biofilm structure and had no significant effect on the obtained results.

Under the same experimental conditions, we tested the efficiency of the appearance of antibiotic-resistant bacteria. Fractions of such resistant cells were calculated among survivors that remained after treatment with phages, antibiotics, or their combinations (according to the scheme of treatments shown in Figure 2. In these analyses, we were not able to detect ciprofloxacin-resistant cells, as all tested bacterial cells which survived the treatments remained sensitive to this antibiotic, suggesting that their viability might be due to protection by biofilm structures, though effects of the presence of persistent cells or slow growth of the bacteria cannot be excluded. However, we were able to perform such analyses with rifampicin-resistant cells that appeared in the course of conducting these experiments. We found again that the most effective treatment, causing the lowest number of antibiotic-resistant cells, was the combination of phages (either vB_Eco4M-7 alone or in the phage cocktail) with antibiotic where phage lysate was applied first, and then followed by the treatment with rifampicin (Figure 4).

Analogously, we have calculated the frequency of appearance of phage-resistant *E. coli* cells among survivors. In this case, we were able to assess the results of the experiments conducted in the presence of both tested antibiotics, ciprofloxacin, and rifampicin. The results were again similar and indicated that the most effective way to weaken the bacteria protected in the biofilm is sequential incubation with bacteriophages (especially in the form of the phage cocktail) and then with antibiotic (either ciprofloxacin or rifampicin). Under such conditions, the efficiency of the appearance of bacteria resistant to phages was the lowest (Figure 5).

It is worth noting that no ciprofloxacin-resistant mutants were detected under any tested conditions, while there were some isolates resistant to both phage and rifampicin, especially in experiments with sequential treatment with the phage cocktail and this antibiotic (34% of survivors were resistant to both agents).

Finally, we tested if the analyzed antibacterial agents (individually or in combination) cause induction of Shiga toxin-converting prophage ST2-8624, present in the genome of the *E. coli* O157:H7 (ST2-8624) strain. It was demonstrated previously that vB_Eco4M-7 did not cause the prophage induction during lytic infection of host bacteria in a one-step growth experiment and lysis profile assays [15]. As a positive control, we used mitomycin C which is known as a potent inducer of lambdoid prophages [29]. All tested antibiotics, mitomycin C, ciprofloxacin, and rifampicin, effectively halted the growth of liquid cultures of the *E. coli* O157:H7 (ST2-8624) strain when added to the final concentrations equal to MIC values (Figure 6A). A more spectacular effect of growth inhibition relative to the control variant was observed when the bacterial cultures were treated with the phage vB_Eco4M-7 (MOI of 0.1) or the phage cocktail (MOI of 0.1), individually or in combination with antibiotics. As expected, mitomycin C caused effective prophage induction as assessed by the detection of infective viral particles (plaque-forming units, PFU) in the antibiotic-treated cultures, in contrast to untreated bacteria (Figure 6B). Ciprofloxacin was also found to be effective in the prophage induction. However, the number of infective ST2-8624 virions per ml was approximately 100 times lower than those observed after treatment of host bacteria with mitomycin C (Figure 6B). However, no Shiga toxin-converting phages could be detected in *E. coli* O157:H7 (ST2-8624) cultures grown in the presence of rifampicin (Figure 6B).

Interestingly, the mixture of the phage cocktail and mitomycin C or ciprofloxacin significantly reduced efficiency of spontaneous induction of ST2-8624 prophage, making the number of phages close to the detection limit (precisely, in some repeats of the experiments either no or a few plaques could be observed). Surprisingly, infection of host cells with the Eco4M-7 phage or the phage cocktail (either individually or in combination with rifampicin), has completely inhibited the process of spontaneous induction of the Stx prophage. Such results confirmed our assumptions that vB_Eco4M7 and the phage cocktail alone or in combination with rifampicin C are not inducers of the Stx prophage present in the *E. coli* O157:H7 (ST2-8624) genome (Figure 6B).

## 3. Discussion

Formation of biofilms can significantly impede treatments of bacterial infections as these structures effectively protect microbial cells against various physical and chemical agents, including antibacterial compounds [12,13]. This hindrance might be especially problematic when infections by hard-to-treat bacteria, like STEC, are considered [11]. Therefore, finding novel ways to combat such bacteria is urgent, through this task is neither easy nor simple.

One of the possible approaches to effectively treat STEC infections is phage therapy [14]. This approach is based on the use of virulent bacteriophages infecting STEC cells which should lead to the elimination of these pathogenic bacteria. This idea has been developed previously, and various attempts of the use of phage therapy in the potential treatment of infections caused by STEC have been reported. They included the use of various STEC-specific phages used alone on in cocktails, as well as in combination with probiotics [30,31,32,33,34,35,36,37,38,39]. Moreover, the use of bacteriophages (without combination with other agents) against a biofilm-forming STEC strain has been reported recently [40]. Despite such numerous attempts, a recent review paper on this subject indicated that the results reported to date are not very encouraging in the light of the available methods to eliminate STEC cells, though it was strongly suggested that more work is necessary and studies on the use of phages against STEC are reasonable in order to find more efficient treatment procedures [41]. Such an opinion was corroborated by another group who suggested that phage therapy might be also useful as a prophylactic treatment for humans endangered by STEC infection [42].

The idea of the use of phage therapy and antibiotics together was previously proposed by several research teams [28,43,44,45,46,47,48,49,50,51,52,53,54,55,56,57,58,59,60,61,62]. When considering such a method to combat STEC, Easwaran et al. [46,63] tested the efficacy of phage EcSw administered in combination with kanamycin, chloramphenicol, and/or ampicillin against *E. coli* O157:H7. The results of that study were encouraging; however, two problems remained unsolved. First, effects on biofilms were not tested, and it was indicated previously that antibiotics may cause induction of Shiga toxin-converting prophages, thus, they should not be used in the treatment of STEC infections [10]. Conditions in biofilms may differ significantly from those occurring in liquid bacterial cultures in means of prophage induction efficiency due to both drastically different bacterial growth rates and different local environments affecting the physiology of microbial cells. Hence, in our studies, we assessed the efficiency of STEC cells elimination and reduction of biofilm density by a combination of phage therapy and different antibiotics, while testing also prophage induction and appearance of antibiotic-resistant and phage-resistant mutants.

The results presented in this report indicated that a combination of phage therapy and antibiotics is effective in lowering the number of STEC bacteria present in biofilm, as well as in decreasing biofilm density, especially if treatment with phages (preferably with the phage cocktail) precedes the application of an antibiotic.

We decided to use the phage as the first agent in the sequential treatment to get rid of the released cells and phages, and to allow the antibiotic to reach deeper layers of the biofilm, and thus enhance its action in this way. Bacteriophages have developed different mechanisms to overcome the high bacteria density within the biofilm. They self-amplify effectively and can reach high concentrations; moreover, they are able to penetrate into deeper layers of the biofilm [64] and may produce enzymes, lysins and depolymerases, that allow them to actively spread, disrupt and propagate within biofilms. Phage lysins act on bacterial cell walls and allow them to destroy bacteria at low metabolic rates [65] whereas depolymerases facilitate the early stages of phage infection by degradation of the exopolysaccharide (EPS) matrix components and reduction of the biofilms viscosity [12,66]. Interestingly, these enzymes were also reported to act in a synergistic manner allowing for even more effective removal of biofilms [67]. Due to their action, phages have easier access to the bacterial cells from the deeper layers of the biofilm than other antibacterial agents, like antibiotics, and hence are recommended to be used before other agents to loosen the biofilm structure and so facilitate their action. Such an effect might be also due to the initial relaxation of the structure of the biofilm by bacteriophages which are able to infect bacteria present in this structure due to penetration of the matrix, using a long tail that works as a “syringe” to allow injection of the phage genetic material into the host cell. Although lengths of phage tails are in the range of hundred(s) nm, these viruses might initially infect host cells located in the upper part of the biofilm, and following propagation in the hosts and lysing such cells, they might migrate deeper and deeper, when infecting adjacent bacteria.

The strategies of different orders of administrating phages and antibiotics to treat bacterial biofilms have been investigated previously. It was reported that the highest eradication of *Pseudomonas aeruginosa* biofilms was achieved when they were treated with phage before antibiotics, and it was proposed that phage may increase the effectiveness of antibiotics [28]. Other findings demonstrated that the order in which *Staphylococcus aureus* biofilms were treated with phage and antibiotics, was the most important determinant of biofilm reduction results [68]. Namely, it was shown that the biofilm pre-treatment with phage enhances the activity of selected antibiotics and ensures the highest reduction of viable cells.

Thus, we decided to use a treatment order in which phages were added first and then replaced with antibiotics that were added to the supra-MIC concentrations. Such an approach was proposed previously [69] due to the fact that the application of MICs against bacterial biofilms was often ineffective [70,71]. This phenomenon is associated probably with the difference in antibiotic susceptibility between planktonic bacterial cells and the biofilm community [71,72].

Analyses of genomes of vB_Eco4-M7 and ECML-117 indicated the absence of genes coding for typical depolymerases [15]. On the other hand, in the genome of phage vB_Eco4-M7, we identified a gene coding for a potential soluble lytic murein transglycosylase [15]. The bioinformatic analysis carried out using Pfam database [73] confirmed the presence of the SLT domain (SLT_2 family) responsible for its potential hydrolytic activity. Therefore, it is plausible that the soluble lytic murein transglycosylase of the vB_Eco4m-7 phage may belong to the murein-degrading enzymes which catalyse the cleavage of the glycosidic bonds between N-acetylmuramic acid and N-acetylglucosamine residues in bacterial peptidoglycan. Although typical depolymerases are able to degrade capsular polysaccharides, lipopolysaccharides, and exopolysaccharides, one group of phage depolymerases are O-glycosyl hydrolases (EC 3.2.1.X), enzymes that catalyse the hydrolysis of glycosidic bonds [74]. Nevertheless, the specific activity and function of the above mentioned vB_Eco4-M7-encoded enzyme remain to be determined. 

In our experiments with biofilms, we have used supra-MIC concentrations of antibiotics (400–500 × MIC). These values were chosen on the basis of previous recommendations [69], as lower amount of these compounds were ineffective in combating biofilms. Such high doses of antibiotics would be perhaps impossible to use in internal treatment of patients, where lower doses should be considered. However, the high doses might be practical in protection of either food or various materials.

The major remaining problem is the use of a proper antibiotic that can be combined with phages in the complex therapy against STEC. A recent comprehensive review article on the effects of different antibiotics on these bacteria, and especially on induction of Stx prophages, indicated that some groups of antibiotics (like β-lactams, trimethoprim-sulfamethoxazole, and fluoroquinolones) are especially effective prophage inducers while others can be considered as compounds unable to induce such prophages [10]. Indeed, we confirmed that mitomycin C and ciprofloxacin were potent inducers of the Stx prophage occurring in the genome of the model STEC strain. On the other hand, we were not able to detect a considerable induction of Stx prophages from the bacterial genome after treatment with rifampicin. These results corroborated the previous finding that rifaximin does not induce prophages in STEC cells [75]. Since rifaximin and rifampicin belong to the same group of antibiotics (rifamycins), it appears that they can be considered as potential anti-STEC drugs used in combination with STEC-specific bacteriophages.

## 4. Materials and Methods

### 4.1. Bacteria, Bacteriophages, Media, and Growth Conditions

The *E. coli* O157:H7 (ST2-8624) strain, bearing the Shiga toxin-converting prophage ST2-8624 (Δ*stx2*::*cat gfp*) [76,77], was the host of choice for phages vB_Eco4M-7 and ECML-117 [15,16,78]. The *E. coli* strain C600 [79] was used as the host for the titration of phage ST2-8624 [77]. The liquid cultures were grown in Luria–Bertani (LB) medium (BioShop, Burlington, ON, Canada) with aeration at 37 °C in a shaking incubator (200 rpm; Benchmark Scientific, Sayreville, NJ, USA) or were plated on a solid LB medium with 1.5% agar (LA medium; BTL, Łódź, Poland). Petri dishes, filled with LA medium, were incubated at 37 °C for 24 h. The top agar containing LB broth and 0.7% agar was used in the double agar layer assay. Biofilm studies were carried out by using the *E. coli* O157:H7 (ST2-8624) strain bearing the pUC18 plasmid and F’ plasmid from *E. coli* ER2738 (this study). These bacteria were grown at 37 °C for 48 h without shaking in a 12-well polystyrene plate (Nest Scientific USA, Woodbridge, VA, USA) filled with M9 medium supplemented with 0.2% glucose (BioShop, Burlington, ON, Canada).

### 4.2. Propagation of Bacteriophages

To obtain lysates of phages vB_Eco4M-7 and ECML-117, a previously described procedure [15] was used. Briefly, the *E. coli* O157:H7 (ST2-8624) strain was grown to an optical density of 0.2 measured at a wavelength of 600 nm. At this stage, the bacterial culture was infected by phage vB-Eco4M-7 or ECML-117 at an M.O.I. of 0.1 and subsequently incubated at 37 °C with shaking (200 rpm). After 2 h of cultivation, the bacterial cell debris was removed by centrifugation (2000× *g*, 10 min, 4 °C). The obtained supernatant with phage particles was passed through the filter with a 0.22-µm-pore size SFCA membrane (MERC, Kenilworth, IL, USA) and then stored at 4 °C for further analyses.

### 4.3. Determination of the Concentration of Phage Particles in a Viral Stock

To determine the phage titer in a viral stock, expressed in the number of plaque-forming units per ml (PFU/mL), the double agar layer assay was used [16]. Serial 10-fold dilutions of phage lysate were prepared in TM buffer (10 mM Tris-HCl, 10 mM MgSO4, pH 7.2; BioShop, Burlington, ON, Canada). In the next step, 0.1 mL of each dilution was mixed with 1 mL of the overnight host culture and 2 mL of the top agar. The mixture was quickly poured onto a Petri dish filled with LA medium. Double agar plates were incubated at 37 °C overnight and then the plaque-forming units were counted.

### 4.4. Determination of the Minimum Inhibitory Concentration (MIC) of Tested Antibiotics

The minimum inhibitory concentrations (MICs) of three antimicrobial agents: mitomycin C (BioShop, Burlington, ON, Canada), ciprofloxacin (MERC, Kenilworth, IL, USA), and rifampicin (BioShop, Burlington, ON, Canada) were determined for *E. coli* O157:H7 (ST2-8624) strain by using microdilution method described earlier [80,81], with some modifications. Briefly, ciprofloxacin and rifampicin were dissolved in LB medium to the final concentrations ranging from 0.00005 µg/mL to 256 µg/mL and transferred to the wells of a 96-well polystyrene plates (Nest Scientific USA, Woodbridge, VA, USA). For mitomycin C, the solutions of concentrations from 0.0002 µg/mL to 125 µg/mL were prepared. In the next step, the mixtures were added to overnight bacterial cultures (inoculum of 3 × 10^6^ colony forming units, CFU). Plates were incubated with aeration at 37 °C in a shaking incubator (200 rpm; Benchmark Scientific, Sayreville, NJ, USA) for 24 h. Grow inhibition kinetics were determined with the Varioscan Lux plate reader (Thermo Fisher Scientific, Waltham, MA, USA) at a wavelength of 600 nm. The MIC was defined as the lowest concentration of tested agents that prevented a considerable growth of *E. coli* O157:H7 (ST2-8624) bacteria.

### 4.5. Biofilm Challenge

The biofilm of *E. coli* (ST2-8624) strain bearing the pUC18 plasmid and F’ plasmid was prepared according to a procedure described earlier [26], with some modifications. Briefly, the culture of host bacteria was diluted in M9 medium with 0.2% glucose to an OD_600_ of 0.04–0.06. Then, 2 mL of bacterial suspension was transferred to each well of the 12-well polystyrene plate (Nest Scientific USA, Woodbridge, VA, USA). After 48 h of the static incubation at 37 °C, the medium was discarded and surface-attached bacteria were washed with 1 mL of PBS buffer to remove planktonic cells (BioShop, Burlington, ON, Canada). Note that our definition for mature biofilm was based on the fact that the bacterial cell densities measured at 600 nm were no longer increasing at 48 h, relative to values measured at 24 h. Then, the biofilm was individually treated with (i) LB medium (control), (ii) ciprofloxacin (15 µg/mL per well, 500 × MIC; MERC, Kenilworth, NJ, USA), (iii) rifampicin (3.2 mg/mL per well, 400 × MIC; MERC, Kenilworth, NJ, USA), (iv) phage vB_Ec4M-7 (10^10^ PFU per well is equal to an MOI of 100) and (v) the phage cocktail (10^10^ PFU per well is equal to an MOI of 100) for additional 6 h. In the case of control wells, the medium was added instead of an antimicrobial agent. The MICs of liquid bacterial culture were used to calculate the supra concentration (supra-MIC) of 400–500 times the MIC values as presented by [34]. Moreover, the efficacy of simultaneous and sequential combinations was tested. In simultaneous combination, the mixture containing one of the selected antibiotics (ciprofloxacin at 500 × MIC or rifampicin at 400 × MIC) and phage vB_Eco4M-7 (10^10^ PFU per well is equal to an MOI of 100) or the phage cocktail (10^10^ PFU per well is equal to an MOI of 100) was added into biofilm-bearing wells and incubated for 6 h at 37 °C without shaking. In sequential combination, vB_Eco4M-7 (10^10^ PFU per well which was equal to an MOI of 100) or the phage cocktail (10^10^ PFU per well which was equal to an MOI of 100) was used against the biofilm for 3 h. Then, the phage-containing medium was gently removed using a simple laboratory pipette (such a technical approach does not disturb the structure of the biofilm) and ciprofloxacin (500 × MIC) or rifampicin (400 × MIC) was applied for an additional 3 h. Following the incubation with antimicrobial agents (individually, sequentially, or simultaneously) the mixture was discarded and the biofilm was washed once with 1 mL of PBS buffer (BioShop, Burlington, ON, Canada) to remove loosely adherent planktonic cells (BioShop, Burlington, ON, Canada).

The antibiofilm activity of tested agents was analysed by measuring the absorbance of bacterial culture and determining the number of survivors per ml of the suspension (colony forming units per ml or CFU/mL) as described earlier [26,82], with some alterations. Therefore, 1 mL of PBS buffer was added to each well, and adherent biofilm cells were disrupted by vigorous pipetting. The optical density of suspended biofilm was measured with the Varioscan Lux plate reader (Thermo Fisher Scientific, Waltham, MA, USA) at a wavelength of 600 nm. In turn, to estimate CFU/mL the 10-fold serial dilutions of suspended biofilm were performed in 0.85% sodium chloride (Chempur, Piekary Śląskie, Poland), and 0.04 mL of each dilution was directly spread onto a LB agar plate. After overnight incubation at 37 °C, the bacterial survivors were counted, and the CFU/mL was determined.

Additionally, crystal violet (CV) staining was used to monitor the disruption of bacterial biofilm by the Eco4M-7 phage or the phage cocktail according to the procedure described earlier [26,82], with minor modification. Briefly, the biofilm layer was air-dried at 37 °C for 15 min and stained with 0.1% crystal violet (MERC, Kenilworth, NJ, USA) for 30 min at room temperature. In the next step, CV was removed and the excess stain was rinsed off by 5 times washing of biofilm with 1 mL of PBS (until PBS buffer was colorless). Then, the biofilm was fixed by incubating the plates at 60 °C for 30 min. To solubilize the dye bound to the biofilm, 1 mL of 96% ethanol (Avantor Performance Materials Poland S.A., Gliwice, Poland) was added to each well, and the optical density at 570 nm was measured in the microplate spectrophotometer.

The biofilm area after fixation was also photographed to indicate the differences between the control variant and the biofilm biomass treated with antimicrobial agents alone or in combination. The biofilm thickness was measured by densitometry, using UVITEC Allience Q9 Mini software as described earlier [26,82], with some modifications. Briefly, the wells with adherent biofilms were digitized, and the images were used for densitometric quantification. The total pixel density for each well was determined by drawing a circle around the bacterial biofilm. The circles were of equal sizes, including one that serves as the background drawn in a blank area of the empty well. The quantification of the pixel density of the defined biofilm areas was quantified after subtraction of the pixel count for the background.

### 4.6. Development of Biofilm Resistance to Tested Antibiotics

To assess the emergence of antibiotic-resistant bacteria among survivors from biofilms treated with antimicrobial agents, bacterial colonies were passaged onto LA plates supplemented with ciprofloxacin or rifampicin to the final concentrations of 0.06 µg/mL or 8 µg/mL, respectively. After overnight incubation at 37 °C, the number of resistant bacteria among survivors was calculated.

### 4.7. Detection of Phage-Resistant Bacteria in a Biofilm

To determine the number of phage-resistant bacteria among survivors recovered after biofilm treatment with tested agents, procedures described earlier [16,83] were used, with some modifications. In the first stage, all survivors were transferred onto the surface of a top agar mixed with indicator host strain [34]. After overnight incubation at 37 °C, all colonies with and without the presence of the lysis zone were regrown in LB medium to OD_600_ of 0.2 at 37 °C. Then, the vB_Eco4M-7 lysate was added to each sample to an MOI (multiplicity of infection) of 0.1. Following overnight incubation at 37 °C, the optical density of bacterial culture was measured with the Varioscan Lux plate reader (Thermo Fisher Scientific, Waltham, MA, USA) at a wavelength of 600 nm. The sensitivity of all tested survivors to the reinfection with the vB_Eco4M-7 bacteriophage was determined on the basis of a significant decrease in the value of the optical density of bacterial culture and the clarity of the LB medium as recommended previously [16,83]. The percentage of phage resistant bacteria was calculated relative to the number of all survivors recovered from the biofilm.

### 4.8. Induction of the ST2-8624 Prophage (Stx Prophage) in Host Cells

To test whether the treatment of E. coli O157:H7 (ST2-8624) strain with antimicrobial agents alone or in combination causes the induction of the prophage ST2-8624, the kinetic of bacterial culture lysis was determined according to the procedures described earlier [15,84], with some modifications. Briefly, host bacteria lysogenic for the phage ST2-8624 were grown in LB medium at 37 °C to OD_600_ of 0.15. Then, the culture was divided into two aliquots. One of them was treated with the tested agent (alone or in combination), while the second one was inoculated with LB medium (the negative control). The vB_Eco4M7 lysate or the phage cocktail were added to the host bacteria to an MOI of 0.1. To provoke the prophage ST2-8624 induction the MIC concentration of mitomycin C, ciprofloxacin and rifampicin were tested. Then, 0.15 mL of the bacterial culture was transferred to a 96-well polystyrene plate (Nest Scientific USA, Woodbridge, VA, USA) and the cultivation was continued at 37 °C for 6 h in the Varioscan Lux plate reader (Thermo Fisher Scientific, Waltham, MA, USA). At indicated times, the density of bacterial culture, monitored by OD_600_ measurement, and the number of phage particles per ml (PFU/mL) were examined. To determine PFU/mL, the samples were harvested, mixed with chloroform, and centrifuged (2000× *g*, 5 min, room temperature). The obtained lysate of phage ST2-8624 was 10-fold diluted in TM buffer (10 mM Tris-HCl, 10 mM MgSO_4_, pH 7.2; BioShop, Burlington, ON, Canada) and titrated according to the procedure described previously [77], with some modifications. Briefly, a volume of 0.04 mL of each dilution of phage lysate was mixed with 1 mL of the overnight culture of the E. coli C600 strain (as it was presented earlier [15], the indicator strain is resistant to both vB_Eco4M7 and ECML-117 phages). Then, the mixture was added to 2 mL of top agar supplemented with 10 mM CaCl_2_ (BioShop, Burlington, ON, Canada) and 10 mM MgSO_4_ (BioShop, Burlington, ON, Canada), and poured onto the bottom agar with the sublethal concentration of chloramphenicol (2.5 µg/mL; BioShop, Burlington, ON, Canada). Supplementation of the LA medium with an appropriate antibiotic was used to obtain visible plaques formed on the bacterial lawn by ST2-8624 phage. After overnight incubation at 37 °C, the phage titer was determined by counting single plaques.

### 4.9. Statistical Analyses

Each experiment was repeated three times and variations among biological replicates were presented as error bars indicating the standard deviation (SD). Technical replicates were averaged to produce replicate means that were used for analysis. Mean values were compared by using the one-way ANOVA followed by Tukey (equal variances) or Games-Howell (unequal variances) post hoc tests. Levene’s test was used to check the homogeneity of variance assumption. The *p* values < 0.05 were considered statistically significant. All statistical analyses were performed using Microsoft^®^ Excel 365 with Real Statistics Resource Pack.

## 5. Conclusions

The use of the phage cocktail, composed of bacteriophages vB_Eco4M-7 and ECML-117, in combination with either ciprofloxacin or rifampicin, revealed a synergistic effect and was effective in reducing biofilms formed by STEC. Since rifampicin did not induce the prophage present in the genome of the investigated STEC strain, this antibiotic or related compounds might be considered as promising anti-STEC agent(s) when used together with the STEC-specific phage cocktail. 

## Figures and Tables

**Figure 1 antibiotics-11-00712-f001:**
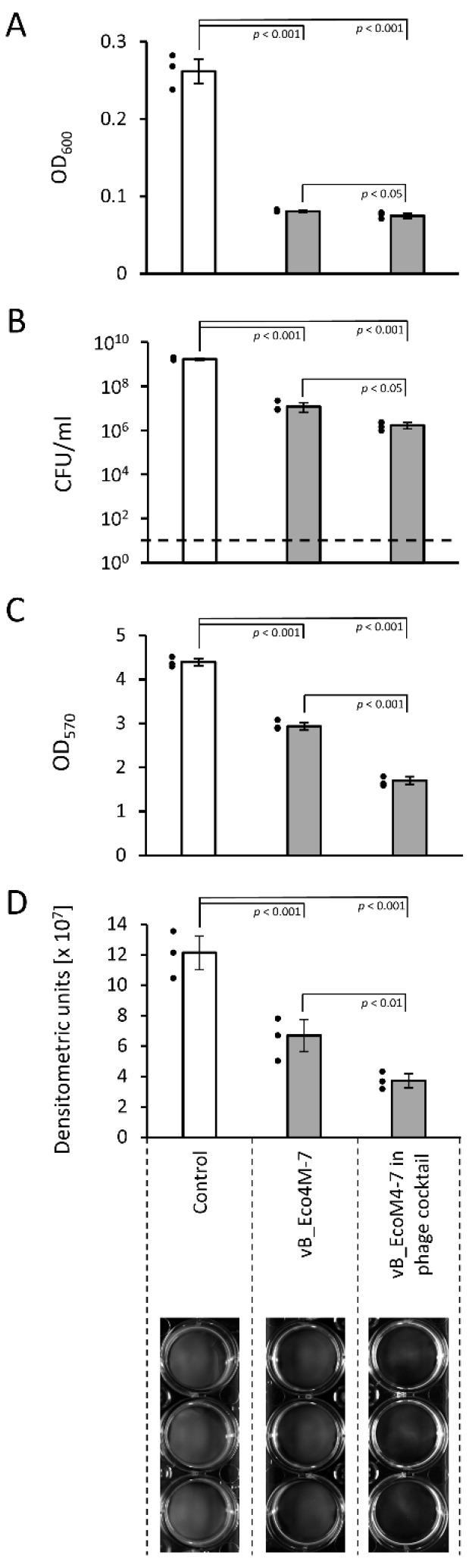
Bacterial biofilm density and viability of cells after a 6-h incubation with the vB_Eco4M-7 phage or the phage cocktail added at an MOI of 100 (10^10^ PFU/well). The results are presented as the bacterial culture density measured at OD_600_ (**A**), the number of survivors per ml of the suspension (CFU/mL) (**B)**, the optical density values measured at wavelength 570 nm after biofilm staining with CV (**C**), or quantified densitometrically using UVITEC Allience Q9 Mini software (**D**). Signatures of the X-axis are the same for all panels and are placed at the bottom of the figure for clarity. The black dashed line represents the lower limit of detection (**C**). Mean values from three independent experiments are shown with error bars indicating SD. The black dots next to the columns denote datapoints from individual experiments. The statistical analyses were performed using the one-way ANOVA (*p* value < 0.05 was considered statistically significant).

**Figure 2 antibiotics-11-00712-f002:**
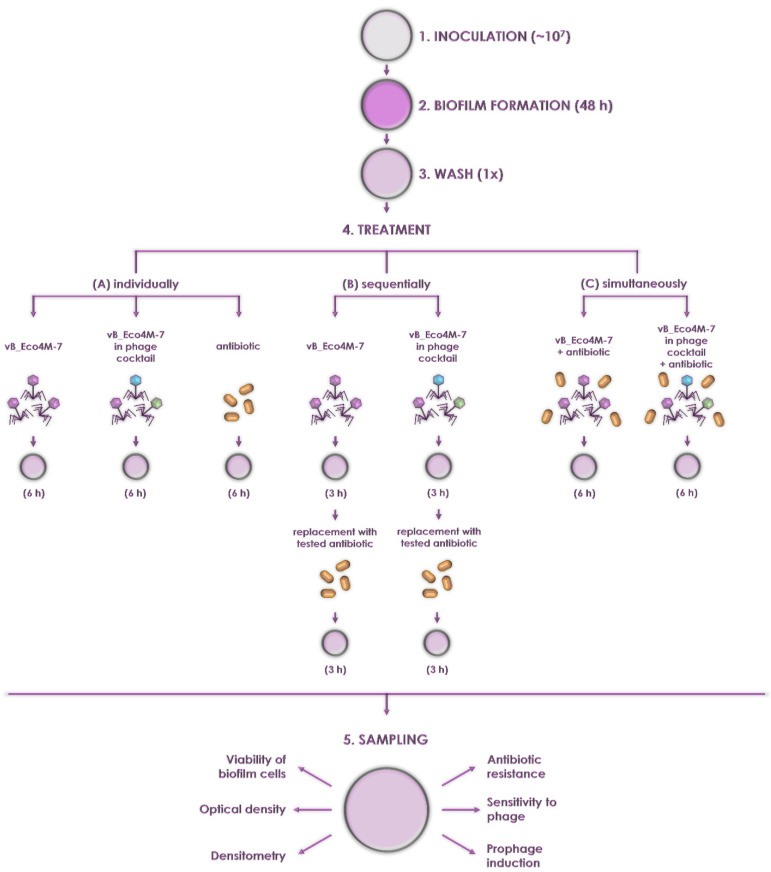
Schematic representation of the procedure used to assess ability of tested antimicrobial agents to eliminate *E. coli* O157:H7 (ST2-8624) bacteria from the biofilm. Steps 1–2 indicate the formation of the biofilm in a 12-well polystyrene plate filled with M9 medium supplemented with 0.2% glucose. Step 3 shows the removal of the planktonic cells and the rinsing of the surface-attached bacteria with PBS. Step 4 presents the eradication of bacterial biofilm with selected antimicrobial agents added individually (**A**), sequentially (**B**), or simultaneously (**C**). Step 5 depicts the analysis of the effectiveness of tested antimicrobial agents using appropriate methods.

**Figure 3 antibiotics-11-00712-f003:**
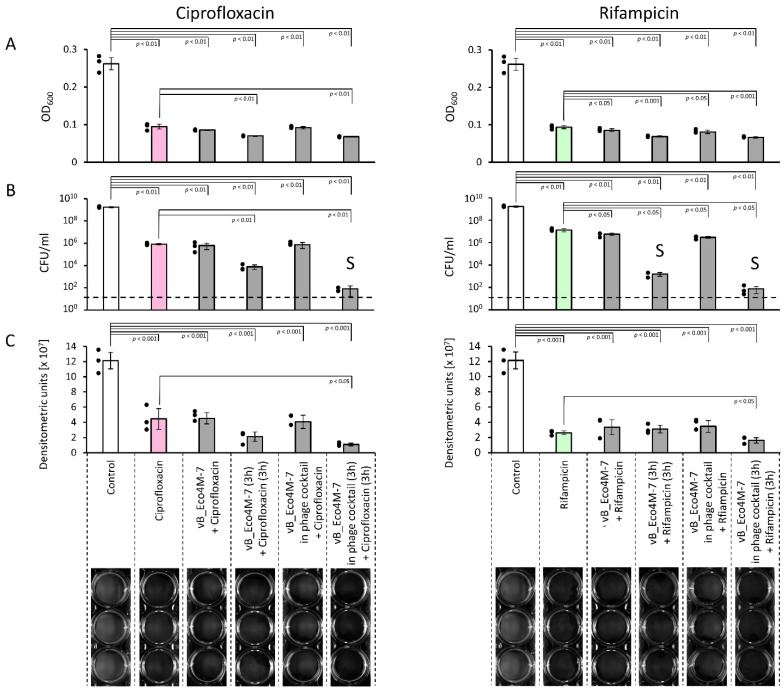
Bacterial biofilm density and viability of cells after a 6-h treatment with different antimicrobial agents individually or in the following combinations: ciprofloxacin (500 × MIC) or rifampicin (400 × MIC); vB_Eco4M-7 phage (MOI of 100) mixed with ciprofloxacin (500 × MIC) or rifampicin (400 × MIC); vB_Eco4M-7 phage (MOI of 100) replaced after a 3-h incubation with ciprofloxacin (500 × MIC) or rifampicin (400 × MIC); the phage cocktail (MOI of 100) mixed with ciprofloxacin (500 × MIC) or rifampicin (400 × MIC); the phage cocktail (MOI of 100) replaced after a 3-h incubation with ciprofloxacin (500 × MIC) or rifampicin (400 × MIC). The results were estimated by measuring the optical density at a wavelength of 600 nm (**A**), enumerating the number of survivors per ml of the suspension (CFU/mL) (**B**), or quantified densitometrically using UVITEC Allience Q9 Mini software (**C**). Signatures of the X-axis are the same for all panels and are placed at the bottom of the figure for clarity. “S” indicates synergistic interaction of tested agents (**B**). The black dashed lines represent the lower limit of detection (**B**). The results are presented as the mean values ± SD from three independent experiments. The black dots next to the columns denote datapoints from individual experiments. Statistical analyses were performed using the one-way ANOVA (*p* value < 0.05 was considered statistically significant).

**Figure 4 antibiotics-11-00712-f004:**
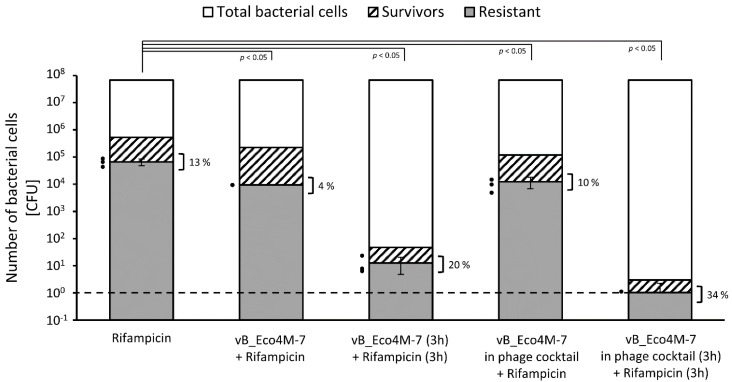
Number of surviving cells (columns with black stripes) and rifampicin-resistant *E. coli* O157:H7 (ST2-862) mutants (gray columns) among survivors after biofilm treatment for 6 h with different antimicrobial agents, individually or in the following combinations: rifampicin (400 × MIC), vB_Eco4M-7 phage (MOI of 100) mixed with rifampicin (400 × MIC), vB_Eco4M-7 phage (MOI of 100) replaced after a 3-h incubation with rifampicin (400 × MIC), the phage cocktail (10^10^ PFU/well) mixed with rifampicin (400 × MIC), the phage cocktail (MOI of 100) replaced after a 3-h incubation with rifampicin (400 × MIC). Control represents the total number of bacterial cells (white columns) after treatment with LB medium instead of tested antimicrobial agents. Additionally, the percentage of rifampicin-resistant mutants (gray columns) among survivors (columns with black stripes) was calculated and presented next to the clamps. The black dashed line represents the lower limit of detection. Mean values from three independent experiments are shown with error bars indicating SD. The black dots next to the columns denote datapoints for rifampicin-resistant mutants from individual experiments. Statistical analyses were performed using the one-way ANOVA (*p* value < 0.05 was considered statistically significant).

**Figure 5 antibiotics-11-00712-f005:**
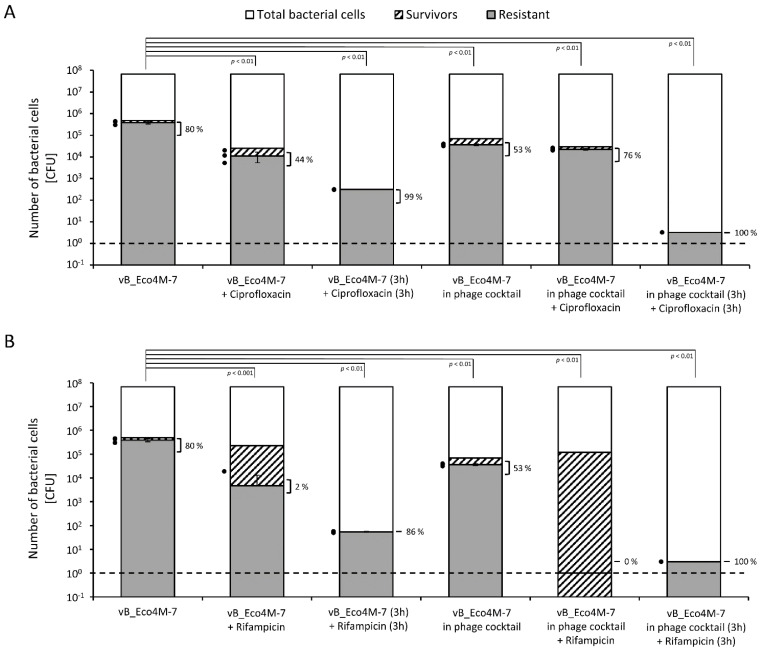
Number of surviving cells (columns with black stripes) and phage-resistant *E. coli* O157:H7 (ST2-862) mutants (gray columns) among survivors after biofilm treatment for 6 h with tested phage, individually or in combinations with ciprofloxacin (**A**) or rifampicin (**B**); vB_Eco4M-7 phage (MOI of 100), vB_Eco4M-7 phage (MOI of 100) mixed with antibiotic, vB_Eco4M-7 phage (MOI of 100) replaced after a 3-h incubation with antibiotic, the phage cocktail (MOI of 100), the phage cocktail (MOI of 100) mixed with antibiotic, the phage cocktail (MOI of 100) replaced after a 3-h incubation with the antibiotic. Control represents the total number of bacterial cells (white columns) after treatment with LB medium instead of tested antimicrobial agents. Additionally, the percentage of phage-resistant mutants (gray columns) among survivors (columns with black stripes) was calculated and presented next to the clamps. The black dashed lines represent the lower limit of detection. Mean values from three independent experiments are shown with error bars indicating SD. The black dots next to the columns denote datapoints for phage-resistant mutants from individual experiments. Statistical analyses were performed using the one-way ANOVA (*p* value < 0.05 was considered statistically significant).

**Figure 6 antibiotics-11-00712-f006:**
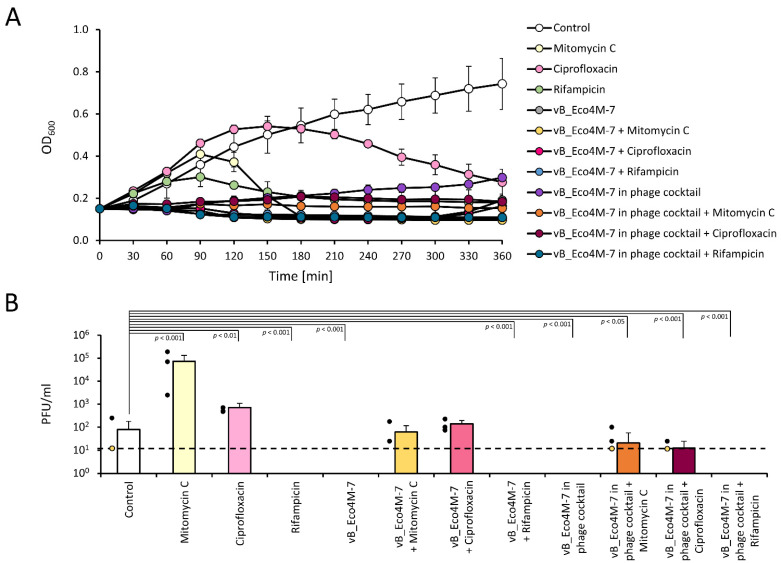
Efficiency of Shiga toxin-converting prophage induction in the *E. coli* O157:H7 (ST2-8624) strain growing in LB medium supplemented with different antimicrobial agents, individually or in the following combinations: mitomycin C (2 µg/mL) or ciprofloxacin (0.03 µg/mL) or rifampicin (8 µg/mL), the vB_Eco4-M7 phage (MOI of 0.1) or the phage cocktail (MOI of 0.1), the vB_Eco4M-7 phage (MOI of 0.1) mixed with mitomycin C (2 µg/mL), ciprofloxacin (0.03 µg/mL) or rifampicin (8 µg/mL), the phage cocktail (MOI of 0.1) mixed with mitomycin C (2 µg/mL) or ciprofloxacin (0.03 µg/mL) or rifampicin (8 µg/mL). Results are shown as bacterial culture density measured at OD_600_ (**A**) and the number of infective phage particles per ml of suspension (PFU/mL) after a 6 h incubation with tested agents (**B**). The black dashed line represents the lower limit of detection (**B**). Mean values from three independent experiments are shown with error bars indicating SD. The black dots next to the columns denote datapoints from individual experiments. Values below the limit of detection (no plaques observed) are marked with yellow dots. Note that in some cases the bars are smaller than the sizes of symbols. Statistical analyses were performed using the one-way ANOVA (*p* value < 0.05 was considered statistically significant).

**Table 1 antibiotics-11-00712-t001:** MIC values (against the *E. coli* O157:H7 (ST2-8624) strain) and mechanisms of action of used antibiotics.

Antibiotic	MIC (g/mL)	Mechanism of Action
Mitomycin C	2	DNA alkylation; it covalently crosslinks DNA, inhibiting DNA synthesis and cell proliferation [22,23]
Ciprofloxacin	0.03	Inhibition of DNA replication by blocking bacterial DNA topoisomerase and DNA gyrase [24]
Rifampicin	8	Inhibition of bacterial DNA-dependent RNA synthesis (transcription) by blocking RNA polymerase [25]

## Data Availability

Raw data are available from authors at request.

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
