# Peer review of "Synergistic Effects of Bacteriophage vB_Eco4-M7 and Selected Antibiotics on the Biofilm Formed by Shiga Toxin-Producing Escherichia coli"

_antibiotics, 2022, doi:10.3390/antibiotics11060712_

Round 1

Reviewer 1 Report

The reviewers satisfactoraly addressed my remaining concern in this resubmission. 

Author Response

REVIEWER’S COMMENT:

The reviewers satisfactoraly addressed my remaining concern in this resubmission. 

RESPONSE:

We thank the reviewer for very useful comments and interesting discussion on the previous stages of the editorial process. They contributed significantly to make our paper considerably better.

Reviewer 2 Report

General comment

Biofilm formation is very important in bacterial pathogenesis and effective strategies for disrupting biofilm formation are limited. This is an interesting investigation on the effect of a previously characterised strictly lytic phage vB_Eco4-M7 specific for E. coli O157:H7 on monoculture biofilms after they were formed. The phage had variable efficacy on cell viability, thickness, and resistance (to antibiotics or phage) in combination with another phage, ciprofloxacin, or rifampicin. The study also examined antibiotic and phage resistance development and prophage induction, both important considerations for phage therapy. While vB_Eco4-M7 in combination with ECML-117 and rifampicin was most efficacious of all combinations tested, some aspects of assay design need more explanation/justification, particularly with regard to feasibility of this therapeutic strategy in animal models/the clinic (e.g. supra-MIC concentrations and physiological toxicity).

Specific comments

Line 93 “..vB_Eco4M7 phage turned out more efficient in killing bacteria…” would be better as “vB_Eco4M7 phage was more efficient in killing bacteria..”

Table 1: Please relate MIC values to antimicrobial susceptibility based on clinical breakpoints for ciprofloxacin and rifampicin. Please also explain why MBC-B values would not be more appropriate given the focus is on biofilm?

Line 117-119. … bacterial cell densities measured at 600 nm were no longer increasing at 48 h..”. I think this belongs in methods, and it needs to be made clear what cell density measure this is relating to. Looking at the methods described for biofilm challenge (line 486 onwards) where OD600 measurements were taken at a few different stages, it wasn’t clear which stage the above is referring to.

Line 143. “..phage cocktail added to an MOI of 100” should be “…phage cocktail added at an MOI of 100”.

Lines 176-202 is not appropriate in results; it should be moved to introduction or discussion.

Lines 202 – 209, including Figure 2 is not appropriate in results, it should be moved to methods.

Line 219 “..highest effectivity in..” probably clearer to say “..highest efficacy in…”

Fig 3. Numbers (1), (2) etc in figure legend do not relate to any corresponding labels in the figure making it hard to follow. Please add numbered labels to the figure for clarity.

The main concern is using supra-MIC, is this feasible in a clinical situation or physiologically relevant? What is the toxicity to the human host under such high doses and can that even be achieved in vivo at a targeted site?

Fig 4 and 5. With regards to assessing resistance to antibiotics or phages, this is an important aspect. Given the strategy, would it not be more relevant to look for cells that are both antibiotic and phage resistant? Or examine whether cells resistant to either phage or antibiotics were susceptible to the other? Could authors clarify whether with the way the assay was designed wouldn’t the resistant cells in Fig 4 and 5 not be both antibiotic and phage resistant?

Fig 6. Nice to see that no prophage induction resulted from vB_Eco4M-7 phage and rifampicin at MIC values. What was the rationale for testing this at MIC values for the antibiotics and MOI of 0.1? Isn’t phage induction usually achieved (in vitro anyway) at sub-MIC values? If cells were unable to replicate under these conditions from the starting culture (as shown in Fig 6A), or the number of viable cells was very low, would that not also follow that no prophage, or no detectable prophage will be induced? Finally, could authors consider whether MIC values are relevant to biofilm? This comment relates to my previous point on determining MBC-B.

Author Response

REVIEWER’S COMMENT:

Line 93 “..vB_Eco4M7 phage turned out more efficient in killing bacteria…” would be better as “vB_Eco4M7 phage was more efficient in killing bacteria..”

RESPONSE:

This was corrected, according to the reviewer’s request (line 93).

REVIEWER’S COMMENT:

Table 1: Please relate MIC values to antimicrobial susceptibility based on clinical breakpoints for ciprofloxacin and rifampicin. Please also explain why MBC-B values would not be more appropriate given the focus is on biofilm?

RESPONSE:

The relation of MIC to clinical breakpoints, and discussion on MBC-B, were included, as suggested by the reviewer. The newly introduced text reads as follows (lines: 109-116):

“The MIC value for ciprofloxacin was significantly lower (0.03 ug/ml) than the clinical breakpoint for this antibiotic in the treatment of Enterobacterales, estimated as 0.25 g/ml; a breakpoint for rifampicin in this group of bacteria has not been established (https://www.eucast.org/fileadmin/src/media/PDFs/EUCAST_files/Breakpoint_tables/v_12.0_Breakpoint_Tables.pdf). We have considered MIC values rather than MBC-B, as inhibition of the bacterial growth appears more important for the practical use (pre-venting multiplication of pathogenic cells) than actual killing of the bacterial cells.”

REVIEWER’S COMMENT:

Line 117-119. … bacterial cell densities measured at 600 nm were no longer increasing at 48 h..”. I think this belongs in methods, and it needs to be made clear what cell density measure this is relating to. Looking at the methods described for biofilm challenge (line 486 onwards) where OD600 measurements were taken at a few different stages, it wasn’t clear which stage the above is referring to.

RESPONSE:

This description has been transferred to the Methods section (lines 509-511). We agree that now, the description is clear.

REVIEWER’S COMMENT:

Line 143. “..phage cocktail added to an MOI of 100” should be “…phage cocktail added at an MOI of 100”.

RESPONSE:

Corrected, as requested by the reviewer (line 150).

REVIEWER’S COMMENT:

Lines 176-202 is not appropriate in results; it should be moved to introduction or discussion.

RESPONSE:

As suggested by the reviewer, this fragment has been transferred to Discussion (lines: 396-418).

REVIEWER’S COMMENT:

Lines 202 – 209, including Figure 2 is not appropriate in results, it should be moved to methods.

RESPONSE:

We agree with the reviewer, that a scheme for the experimental design is generally appropriate for the Methods section. However, this journal requires the Methods section after Results and Discussion. Therefore, putting the scheme of the experiments after showing results and discusion would be useless for a reader. Therefore, we would like to keep Figure 2 where it is, to allow readers easy analysis of the results.

REVIEWER’S COMMENT:

Line 219 “..highest effectivity in..” probably clearer to say “..highest efficacy in…”

RESPONSE:

Corrected, as requested by the reviewer (line 187).

REVIEWER’S COMMENT:

Fig 3. Numbers (1), (2) etc in figure legend do not relate to any corresponding labels in the figure making it hard to follow. Please add numbered labels to the figure for clarity.

RESPONSE:

We thank the reviewer for pointing this discrepancy. In fact, the Figure is complicated enough, thus, instead of adding next numbers to the figure, those were removed from the legend, as the description is still clear without such numbers.

REVIEWER’S COMMENT:

The main concern is using supra-MIC, is this feasible in a clinical situation or physiologically relevant? What is the toxicity to the human host under such high doses and can that even be achieved in vivo at a targeted site?

RESPONSE:

This problem has been discussed now. The following text has been added to Discussion (lines: 432-437):

“In our experiments with biofilms, we have used supra-MIC concentrations of an-tibiotics (400-500 x MIC). These values were chosen on the basis of previous recom-mendations [69], as lower amount of these compounds were ineffective in combating biofilms. Such high doses of antibiotics would be perhaps impossible to use in internal treatment of patients, where lower doses should be considered. However, the high doses might be practical in protection of either food or various materials.”

REVIEWER’S COMMENT:

Fig 4 and 5. With regards to assessing resistance to antibiotics or phages, this is an important aspect. Given the strategy, would it not be more relevant to look for cells that are both antibiotic and phage resistant? Or examine whether cells resistant to either phage or antibiotics were susceptible to the other? Could authors clarify whether with the way the assay was designed wouldn’t the resistant cells in Fig 4 and 5 not be both antibiotic and phage resistant?

RESPONSE:

We thank the reviewer for indicating this important issue. Yes, we found isolated resistant to both phage and antibiotic in the case of experiments with the phage cocktail and rifampicin (but not in experiments with any combinations with ciprofloxacin). This is now described in the text as follows (lines 267-270):

“It is worth to note that no ciprofloxacin-resistant mutants were detected under any tested conditions, while there were some isolates resistant to both phage and ri-fampicin, especially in experiments with sequential treatment with the phage cocktail and this antibiotic (34% of survivors were resistant to both agents).”

REVIEWER’S COMMENT:

Fig 6. Nice to see that no prophage induction resulted from vB_Eco4M-7 phage and rifampicin at MIC values. What was the rationale for testing this at MIC values for the antibiotics and MOI of 0.1? Isn’t phage induction usually achieved (in vitro anyway) at sub-MIC values? If cells were unable to replicate under these conditions from the starting culture (as shown in Fig 6A), or the number of viable cells was very low, would that not also follow that no prophage, or no detectable prophage will be induced? Finally, could authors consider whether MIC values are relevant to biofilm? This comment relates to my previous point on determining MBC-B.

RESPONSE:

It is known that Shiga toxin-encoding prophages can be induced by antibiotics, and effectively replicate in the host cell before it is killed by such a drug (for a review, see Future Microbiol. 2011, 6, 909–924; DOI:10.2217/fmb.11.70). This is why the use of antibiotics is forbidden in many countries if infection with STEC is confirmed or even suspected. Induced prophages can enter the lytic development and produce progeny phages in a relatively short time (20-40 min), before the cell metabolism is completely halted by the action of antibiotic. Therefore, the use of MIC values should not prevent prophage induction and production of progeny phages. This is also why MIC was more useful than MBC-B in this study, as using MBC-B might cause quick killing of bacterial host and block prophage induction, indeed.

This manuscript is a resubmission of an earlier submission. The following is a list of the peer review reports and author responses from that submission.

Round 1

Reviewer 1 Report

In this manuscript, the authors combined phages and antibiotics into a procedure that effectively kills Shiga toxin-producing E. coli (STEC) in biofilm. They found that the vB_Eco4M-7 phage significantly reduced the number of STEC cells in biofilm, which could be further enhanced by using a phage cocktail. Next, they tested various ways of applying antibiotics and phage cocktails individually, sequentially, or simultaneously. Their results suggest that treatment with phages before antibiotics has the optimal outcome. Also, they found that rifampicin did not induce prophages in STEC and could be a good candidate to be used together with the phage cocktails. I enjoy reading this manuscript as it is well-written and easy to follow. The introduction really helps prepare the readers for the following experimental results and discussions. That being said, a few points need to be addressed before publication.

  1. Line 91: please add a paragraph to describe the results in Fig. 1 in detail.
  2. 1: please consider adding the names of the antibiotics to the upper right corner of each panel to help the readers get better visualization.
  3. Line 109: Some brief details about the culture conditions (static or shaking) and measurements (how to get ODs from biofilm samples) would save the readers from going back and forth between this paragraph and the methods section. Also, please add one sentence to help readers understand how densitometric measurement works.
  4. 3: the authors should include data from 1) antibiotics only for 6 h, and 2) antibiotics 3 h -> phage 3 h.
  5. 4 and Fig. 5 could be combined into one for easier comparison.
  6. 6: “Resistant bacterial among survivors” seems incorrect for the context of the figure. Why would this parameter have a unit of CFU/ml? Also, the authors should report the ratio of “survivor cells / total cells” or consider using these to replace CFU bar graphs.
  7. All figures involving CFU or PFU should report the lower limit of detection (especially for Fig. 8).
  8. Line 264-275: this is a rather long list of examples with excessive details. Please consider replacing it with a concise summary.
  9. Line 282-286: It seems unnecessary to repeat what was described in previous sections.
  10. Line 298: “eradication” seems inaccurate since none of the data showed true eradication.
  11. Line 440-443: please provide more details on the statistical analyses.

Reviewer 2 Report

In this manuscript, Necel et al test the efficacy of a phage, phage cocktail, and antibiotics as bactericidal tools, with a particular focus on the ability to affect biofilms.

The manuscript is challenging to review, because it's not clear what the central message is. 

If it is that phage vB_Eco4-M7 is an effective anti-STEC tool, then there are some controls lacking regarding phage ECML-117, which is never used alone and could be responsible for most of the effect. If it is that these phages are effective at degrading biofilms, there is an entire discussion on phage depolymerases that is missing - and likely some experiments on biofilms of different ages (6 h is *very* young), posisbly infections on stationary phase lawns, and maybe direct evidence of biofilm degradation. If it is that there is synergy between phage and antibiotics, there is a large field dating back to ~2007, almost none of which is cited, and there would likely be an expectation of *some* mechanistic insight; is this RecA dependent? If it is that the timing of the phage and antibiotic administraiton matters, again, there are some timing controls that are lacking; antibiotic for 6h, or antibiotic first. If it is that antibiotics and phages can be used and not induce STEC stx-prophages, it seems to me there is a need to demonstrate not just phage creation, but STX levels (either qPCR, reporter assay, or a detection of the toxin directly). If it is that the phage and antibiotic do not select for resistance in the same way as phage-alone or antibiotic alone challenges, there are again some experiments missing to show this - especially as they pertain to the two phages used in combination. And if it is a proposed clinical model, the use of MitC, Cipro and Rif - none of which would ever be prescribed for an STEC infection - is pretty glaring. 

Ultimately, the manuscript doesn't have to be all of these things - or even many of these, but it has to be clear which of these it is targetting and be revised accordingly. This is further aggravated by the fact that the biofilm experiments are not well described in the Results, such that it is unclear how much of the effects (e.g. on cfu) is due to reduction in planktonic cells, how much of the densitrometry has to do with *interfering* with further biofilm formation over the 6 h rather than degradation,e tc. This makes it easy to state that the manuscript should be rejected and revised with further experiments, but challenging to help the authors by identifying which experiments to do - taht depends very much on the narrative they are attempting to complete. 

Major:
All figures do not display individual data points, and only show averages. Please remedy. In most cases, the statistical test used (t test) is not the appropriate test - and the choice of null hypothesis (e.g. Fig 3,4) seems to be the antibiotic alone? Why is this?

Touched on this above, but the controls in Fig 1, 3,4 probably need a time series - control prior to 6 h vs control after 6 h, to distinguish between a reduction in biofilm, and a pprevention of the generation of MORE growth. 

Figure 3 is a nice illustration, but it also highlights so many of the missing components mentioned in my overview; where is the 6 h antibiotic? Where is the second phaeg alone? Where is the 3 h phage alone? Where are the antibitoic-first/then phage conditions? Also notable; the wash is not specifically justified (not sure why the planktonic cells have to go), but I also *really* dont understand why the sequential treatments involve *replacement* rather than addition. 

The choice of extremely high antibiotic concentrations - 500 x MIC! is not really justified for the assays. This is a big departure. Its also problematic because of Fig 8, which is done at MIC - does 500 x MIC completely abolish phage production? It might!

Fig 6 and 7 were very challenging to grasp, because of the log scale and % used - and the awkward use of T tests. May benefit from the % actually being inside the box it refers to (in this case, the resistant). Other major issues; definition of resistance not established? Clinical standard for Europe? US? What ab out intermediate sensitivity? Not clear if all survivors were tested. Not clear why non-sensitive is used for Fig 7 instead of resistant, what is the threshold in EOP for resistance, and in Fig 7, no discussion of the second phage (resistance to it? cross-resistance) is made - this is problematic. FIg 7 B cocktail_Rif is a *big* difference - massive, and could justify a paper in and of itself, if verified correctly. This almost looks like antagonism of some kind, where the Rif is blocking all phage-based selection (or failure to add phage?)

Fig 8 as mentioned earlier, problematic that this is at MIC when your strategy is 500x MIC for effect before. Also problematic that phage+antibiotic conditions are not tested here - just because the phage alone doesnt show induction in previous tests (at same MOI? In biofilms? Unclear) doesnt mean phage+rif, or phage+cipro would have the same effect. I am surprised to see such a low spontaneous induction of the phage - this is FAR below most lambdoid phages that would be expected at 10^5 ish pfu/ml int he absence of a stressor. 

Moderate: 
There is not enough context provided for biofilms. This is true in the intro; (planktonic vs persisiters), but is problematic too because there are existing gold-standard methods for biofilm detection, quantification, etc - CV staining, microscopy, etc, so its not clear why these are eschewed, and it is very challenging to read Fig 2,3,4 without having combed the methods in detail to understand exactly what the OD represents (of the biofilm? Broken up? Resuspended), what the CFU represents (again, of the broken up biofilm), and so on. L106-111 has to be expanded upon (how long the biofilm, how much phage lysate - what MOI, what are you measuring OD600 on, why densitometric )

Fig 1 doesnt warrant a figure - table 1 is sufficient. Fig 1 could be supplemental. 

Fig 1, 3, 4 could easily be combined. 

The term synergy is used throughout, but is problematic. To show synergy, the authors must establish something beyond the additive effect of the two components. 

L299-304 this is pure speculation, and problematic. Biofilms are dozens to hundreds of cells deep - unless htese phages have 100 uM tails (They do not!), this explanation is unsubstantiated. No discussion of phage depolymerasesa?

Justification for 10^10 phages/well,a nd what effective MOI this is is necessary. 

Minor:
Abstract lacking in details (effective? preceded? how effective? What timing)

Reveal is used throughout, usually instead of displays?

In figure 1, the name of the antibiotic appears nowhere in the figure - could replace the A/B/C, or be listed under the X axis.  

The word estimated is used throughout instead of calculated.

L186: not enough justification of biofilm broken up, but also not enough justification that survival has to do with biofilm structures and not persister cells, slow growth, etc. 

L288: This 2020 paper is far from the first - 2007 papers discuss phage antibiotic synergy, and the concept is conceptually older than that even. 

L419: Why is hthis done in broth? EOP by spot test would offer so much more resolution. 

Line by Line:
L26: Tread should be treat
L30: Rif was not introduced earlier in abstract, comes out of nowhere
L37: Run=on-sentence
L45: What about spontaneous induction. Is there no stx transcription in the absence of induction?
L49: What is should be which is?
L82: Three - only two mentioned to date. 
L170 ; Fig 4 and 5 could specific the antibiotic in their title. 
L183: who remained viable can be deleted. 
L260: space missing between effectively and treat. 
L276: Eradication comment is strange?
L292: Any reason to think this induction would be different in biofilms?
L312: INduces should be inducers.
L319: But nobody, nobody uses Rif for E. coli.

Round 2

Reviewer 1 Report

The manuscript has been updated and improved significantly. I think the paper is ready for publication after the authors discuss explicitly in the text the figures where some bars are close to/below the limit of detection.

Reviewer 2 Report

I thank the authors for their revised manuscript, which took the time to address my concerns in considerable detail, in some cases including additional data I found very reassuring. The additional description and emphasis on the biofilms helped focus the manuscript, as did some of the justifications for the avenues explored. 

Critical: No critical concerns

Major: I have only one concern that likely requires additional experiments. This stems from the fact that the sequential treatments include a "replacement with tested antibiotic" step where the replacement itself could be causing some of the effect - especially on the planktonic cells. The methodology used in the "replacement" isn't clear to me, and should be elaborated upon in the methods, but I also worry that the was step in question needs an adequate control in the non-sequential treatments (abx, etc). I will admit that despite the justifications, I do not understand the decision to wash out the previous treatment - certainly in-vivo, these would just be additive by default (add phage, add antibiotics later) rather than (add phage, remove circulating phage, add antibiotics). I think this also ties into a secondary question (not as major) ; do the authors really believe 'synergy' is occuring? It seems to be happening at the level of CFU, but not densitometry or OD - does that reflect a real synergy? Does it reflect simply density-dependent antibiotic sensitivity? 

Moderate:
While this does not require additional experiments, I must re-state the need for individual datapoints within the bar graphs provided. Each replicate is an independent pool of biological events and treating them as technical replicates (normalizing for measurement error) isn't appropriate. The extra effort to display the means from each replicate is worthwhile. 

Some concern in Figs 4 and 5; for instance, is the number of antibiotic resistant cells really the relevant metric, or the frequency relative to the number of survivors? I suppose a case can be made for the former... but if the argument is that the synergy is making frequency less likely *even among survivors* (which one would assume) - in other words, that there is an effect beyond just the increased killing, then I think the focus has to be on the frequency of resistance within survivors - and I'm not convinced these are different across any of those shown in Figure 4. I also believe that the eye-popping before-last column in Fig 5 b warrants a discussion of its own.

The revised data on induction levels is very nice, but it wasn't clear from the methods that the indicator strain was sensitive only to the STX prophage and not any of the cocktail phages. The supression of spontaneous induciton is very cool - I like these data. The language is a little loose in 317-322 and could be tightened to more clearly describe the data in the figure. 

Minor: 
The additional discussion of depoymerases is warranted... but is there any evidence your phages - either of them - have such genes?

I wonder if the synergy/order effects (including those in Ref 32) have more to do with density-dependent antibiotic sensitivity. It's well known that bacterial culture density impacts MIC; a phage challenge reducing bacterial density would make the culture more sensitive to the antibiotic simply by virtue of there being fewer cells.

Unclear why no treatment controsl are not showing up statistically significant in Fig 3.

Line-by-Line
The revised manuscript does have a couple of *massive* paragraphs (L113-141) (161-204) which could be split to facilitate readability. 
L157: Not clear what statistical test was used to determine "significance" here. 
L177: Is the argument that phage make their way through water channels, but in some way antibiotics don't? I think that is unlikely.

Round 3

Reviewer 2 Report

I still have a remaining point that must be cleared up - it was part of my "major" point before, and it was not addressed. I will try to be more explicit.

1) It is not clear to me what method is used to "replace the phage with the antibiotic" (Fig 2B). The "replacement" is not explicitly described in the methods either. 

2) If the replacement is what I suspect (a pelleting and wash, or even a wash) then there should be controls for this wash step, which may reduce biofilm density/planktonic CFU/etc in and of itself (admitedly, wash unlikely to to do much to the biofilm initially, but if the argument is that the phage can break up the b iofilm and sensitize if to the antibiotic, then the wash step is washing away broken-up biofilm as well, and the 'synergy' is in part due to the interaction of the phage with the wash/replacement step. 

In other words, I think there is an important control missing. As such, I still believe it is correct to say "major revisions", even if this is my only remaining point.